# Strategy for Use of Genome-Wide Non-Invasive Prenatal Testing for Rare Autosomal Aneuploidies and Unbalanced Structural Chromosomal Anomalies

**DOI:** 10.3390/jcm9082466

**Published:** 2020-08-01

**Authors:** Pascale Kleinfinger, Laurence Lohmann, Armelle Luscan, Detlef Trost, Laurent Bidat, Véronique Debarge, Vanina Castaigne, Marie-Victoire Senat, Marie-Pierre Brechard, Lucie Guilbaud, Gwenaël Le Guyader, Véronique Satre, Hélène Laurichesse Delmas, Hakima Lallaoui, Marie-Christine Manca-Pellissier, Aicha Boughalem, Mylene Valduga, Farah Hodeib, Alexandra Benachi, Jean Marc Costa

**Affiliations:** 1Laboratoire CERBA, 7/11 Rue de l’Équerre, 95310 Saint-Ouen-l’Aumône, France; llohmann@lab-cerba.com (L.L.); armelle.luscan@lab-cerba.com (A.L.); dtrost@lab-cerba.com (D.T.); Aicha.Boughalem@lab-cerba.com (A.B.); Mylene.Valduga@lab-cerba.com (M.V.); Farah.Hodeib@lab-cerba.com (F.H.); jmcosta@lab-cerba.com (J.M.C.); 2Gynécologie-Obstétrique, Centre Hospitalier René Dubos, 6 av de l’Ile de France, 95300 Pontoise, France; lbidat@wanadoo.fr; 3Gynécologie-Obstétrique, CHU Lille, 2 av Oscar Labret, 59000 Lille, France; veronique.debarge@chru-lille.fr; 4Hopital intercommunal de Creteil, 40 Avenue De Verdun, 94000 Creteil, France; Vanina.Castaigne@chicreteil.fr; 5Medical Department, Université Paris Saclay, 63 Rue Gabriel Péri, 94270 Le Kremlin-Bicêtre, France; marie-victoire.senat@aphp.fr (M.-V.S.); alexandra.benachi@aphp.fr (A.B.); 6Gynécologie-Obstétrique, Hôpital Bicêtre, 78 Rue du Général Leclerc, 94270 Le Kremlin-Bicêtre, France; 7Cytogénétique, Hopital Saint Joseph, 26 Boulevard de Louvain, 13008 Marseille, France; mbrechard@hopital-saint-joseph.fr; 8Service de Médecine Foetale, Hopital Armand Trousseau, APHP Sorbonne Université hôpital Trousseau, 26 Avenue du Dr Arnold Netter, 75012 Paris, France; lucie.guilbaud@gmail.com; 9Génétique médicale, CHU de Poitiers, 2 rue de la Milétrie, CEDEX, CS 90577, 86021 Poitiers, France; gwenael.leguyader@chu-poitiers.fr; 10Génétique Chromosomique, CHU Grenoble Alpes, Avenue Maquis du Grésivaudan, 38700 La Tronche, France; vsatre@chu-grenoble.fr; 11INSERM U1209, CNRS UMR 5309, Institute for Advanced Biosciences, Team Genetics Epigenetics and Therapies of Infertility, Univ. Grenoble Alpes, Avenue Maquis du Grésivaudan, 38700 La Tronche, France; 12Gynécologie-Obstétrique, CHU Clermont Ferrand, 1 Place Lucie et Raymond Aubrac, 63003 Clermont Ferrand, France; helaurichesse@chu-clermontferrand.fr; 13Cylab, 6 Rue des sports BP 60348, CEDEX 1, 17001 La Rochelle, France; hlallaoui@cylab.fr; 14Centre de diagnostic prénatal, Hôpital des enfants de la Timone, 264 Rue Saint-Pierre, 13005 Marseille, France; manca-pellissier@ap-hm.fr; 15Gynécologie-Obstétrique, Hôpital Antoine Béclère, AP-HP, 157 Rue de la Porte de Trivaux, 92140 Clamart, France

**Keywords:** non-invasive prenatal test, genome-wide screening strategy, atypical chromosomal anomalies, rare autosomal aneuploidy, structural unbalanced anomalies, deletion, duplication, sensitivity, specificity, positive predictive value, VeriSeq NIPT Solution v2

## Abstract

Atypical fetal chromosomal anomalies are more frequent than previously recognized and can affect fetal development. We propose a screening strategy for a genome-wide non-invasive prenatal test (NIPT) to detect these atypical chromosomal anomalies (ACAs). Two sample cohorts were tested. Assay performances were determined using Cohort A, which consisted of 192 biobanked plasma samples—42 with ACAs, and 150 without. The rate of additional invasive diagnostic procedures was determined using Cohort B, which consisted of 3097 pregnant women referred for routine NIPT. Of the 192 samples in Cohort A, there were four initial test failures and six discordant calls; overall sensitivity was 88.1% (37/42; CI 75.00–94.81) and specificity was 99.3% (145/146; CI 96.22–99.88). In Cohort B, there were 90 first-pass failures (2.9%). The rate of positive results indicating an anomaly was 1.2% (36/3007) and 0.57% (17/3007) when limited to significant unbalanced chromosomal anomalies and trisomies 8, 9, 12, 14, 15, 16, and 22. These results show that genome-wide NIPT can screen for ACAs with an acceptable sensitivity and a small increase in invasive testing, particularly for women with increased risk following maternal serum screening and by limiting screening to structural anomalies and the most clinically meaningful trisomies.

## 1. Introduction

Several tests are available to screen for chromosomal abnormalities. The most common are first- or second-trimester traditional maternal serum screenings (MSS), which typically screen for common fetal aneuploidies (trisomy 21, 18, and 13) and provide a risk score to determine whether further testing is needed. It has been shown that the risk of atypical chromosomal anomalies (ACAs), that is, rare autosomal aneuploidies (RAAs) or structural unbalanced anomalies (SUAs), increases with the MSS risk score [1,2,3]. Ultrasound anomalies, including nuchal translucency measured between 11 and 13 weeks of gestation, allow detection of some chromosomal anomalies. More recently, non-invasive prenatal testing (NIPT) has become available. NIPT can be performed at any stage of the pregnancy, generally from 10 weeks’ gestation onwards to ensure sufficient fetal fraction (FF) in the maternal plasma sample [4]. NIPT is typically used to screen for common trisomies. Genome-wide screening for ACAs is also possible with NIPT assays using whole-genome sequencing (WGS) and analysis [5,6]. Invasive procedures, namely, chorionic villus sampling (CVS) and amniocentesis to perform a fetal karyotype or microarray are considered diagnostic, but may have a small risk of miscarriage [7,8], as well as a risk of premature rupture of the membranes and chorioamnionitis [9,10].

Clinical studies have shown improved NIPT performance over traditional MSS approaches [11]. Several different technologies are typically used in NIPT assays, primarily massively parallel WGS [12,13,14], targeted microarray hybridization [15], and SNP-based sequencing [16,17]. Massively parallel sequencing techniques use single-end or paired-end sequencing; paired-end sequencing allows determination of both fragment size and location [18,19]. NIPT performance can differ between different methods; a recent meta-analysis [20] reported that NIPT failure rates ranged from 0.1 to 6.3% between methods [21,22,23].

In recent years, test menu options have expanded to include sex chromosome aneuploidies [24,25], select microdeletions/duplications [26,27], and ACAs [5,6,28]. Although ACAs are rarer, NIPT studies have shown a screen-positive rate of 0.1% for SUAs and 0.34% for RAAs [29]. A 2017 study by Fiorentino et al. [5] found that genome-wide cell-free DNA analysis in a large cohort of pregnant women identified clinically relevant imbalances that were not detectable by conventional NIPT while maintaining high specificity. A study by Pescia et al. [30] also found that the use of genome-wide NIPT screening led to an increase in the detection of fetal anomalies, and that the rare autosomal trisomies and CNVs identified through this screen were significant causes of fetal pathology. In France, these ACAs account for 12.25% of all prenatal unbalanced chromosomal abnormalities and 2.5% of all prenatal karyotypes [31]. These ACAs have been shown to be associated with a poor prognosis, including miscarriage, intrauterine fetal death, intellectual development disorders, and malformative syndromes [29,32,33,34,35].

In France, NIPT is reimbursed for trisomy 21 screening only. NIPT is performed as a second-tier screen following a MSS risk score between 1/51 and 1/1000, and as a first-tier screen for women with multiple gestations, history of a previous trisomy 21-affected pregnancy, or where either parent is a known carrier of a Robertsonian translocation involving chromosome 21. A diagnostic test is recommended when there are ultrasound anomalies or the MSS risk score is >1/50.

The objectives of this study were (1) to determine the performance of genome-wide cell-free DNA analysis by the VeriSeq^TM^ NIPT Solution v2 assay for the detection of ACAs, and (2) to determine a strategy for genome-wide NIPT screening for ACAs. Our study found that this genome-wide NIPT assay had high accuracy for the detection of rare autosomal aneuploidies and segmental aneuploidies ≥7 Mb, with a minimal increase in additional confirmatory invasive tests. We also propose a strategy where genome-wide screening is limited to the most significant chromosome trisomies and SUAs, and also offered to women with increased risk following maternal serum screening.

## 2. Experimental Section

This study included two study cohorts. Cohort A consisted of a biobank of plasma samples that were collected to determinate the sensitivity and the specificity of the genome-wide NIPT assay. Abnormal samples were collected between 2014 and 2019 from patients who had an invasive procedure and where karyotype or a microarray (CMA) revealed an anomaly other than a common trisomy (trisomy 21, 18, or 13) or a sex chromosome aneuploidy (Appendix A). All patients with an abnormal karyotype/microarray were offered to participate in the study without any selection based on anomaly type, indication for the invasive procedure, or gestational age. This group represents an unselected group of the chromosomal anomalies that we encounter in our cytogenetic laboratories. The control NIPT samples in Cohort A consisted of the first unselected 148 samples collected from April to July of 2019 that had a karyotype/microarray result without any anomaly other than a common trisomy or sex chromosomal aneuploidy. NIPT samples were collected at the same time as the invasive procedure. Two supplementary NIPT samples were collected after the karyotype result because of a diagnosis of mosaicism confined to the placenta (trophoblast) with a normal karyotype on amniotic fluid.

Cohort B, further called the “referral population”, was collected with the aim of determining the rate of potential additional invasive diagnostic procedures following the use of genome-wide NIPT. It consisted of unselected samples from patients undergoing routine NIPT for common trisomies as part of their clinical work-up. Genome-wide NIPT analysis was subsequently performed for this study. As patients consented to a non-interventional study only, results of the genome-wide analysis were not reported, and we did not contact them to obtain further follow-up information regarding fetal karyotypes and clinical outcomes. These samples were collected between March and August 2019. All samples in both cohorts were from singleton pregnancies.

For Cohort A, karyotyping was performed using standard techniques: in situ culture (Amniomax^®^) or trypsine culture (Amniomedium 2^®^ for amniotic fluid or Amniomax^®^ for trophoblastic culture), hypotonic shock (Hanks^®^ and MgCl_2_), fixation with acetic acid/methanol, and RHG banding. At least 15 metaphases on two cultures were analyzed. Microarrays were performed with the genome-wide array Cytoscan^®^ 750K (SNP Affymetrix, 750K markers) according to the Affymetrix protocol.

All samples were tested with the CE-IVD marked VeriSeq NIPT Solution v2 assay (Illumina, Inc., San Diego, CA, USA). This assay uses a PCR-free, paired-end WGS approach for detection of genome-wide anomalies. Briefly, as plasma samples from both the biobank and the referral population had been frozen, plasma was defrosted followed by a cell-free (cf) DNA extraction step. Purified cfDNA fragments underwent automated library preparation using the VeriSeq NIPT Microlab STAR system (Hamilton, Switzerland), followed by quantification and library pooling. Pooled libraries were sequenced on a NextSeq 550Dx sequencer (Illumina, Inc.). The biobanked samples were tested on the same flow cells used for the samples of the referral population, with a maximum of four biobank samples per flow cell. Bioinformatics analysis was carried out using the VeriSeq NIPT Solution v2 system server, and samples were classified as anomaly detected or anomaly not detected for presence of RAAs or SUAs ≥7 Mb. The assay software uses a dynamic threshold metric known as the individualized Fetal Aneuploidy Confidence Test (iFACT), which takes into account both fetal fraction and coverage information to determine whether a call can be made or not, allowing accurate calls at low fetal fractions.

Comparisons of fetal fractions and gestational age between different patient populations were performed using a *t*-test; a *p*-value of <0.05 was considered significant. Binomial 95% confidence intervals (CI) were calculated for performance estimates.

The study was approved by the “Comité de Protection des Personnes (N° PP 14-007)”. Women gave written consent that their samples could be used for research and that results from this research would not be communicated.

## 3. Results

Cohort A, the biobank samples, consisted of a total of 192 samples (Figure 1). In 42 of the pregnancies, an RAA or SUA ≥7 Mb was found at invasive testing (Appendix A): four RAAs (three trisomy 16 confined to the placenta (type III, i.e., found in both the cytotrophoblast and in the mesenchyme) and one non-mosaic trisomy 22), 32 SUAs (duplication, deletion, or translocation derivative chromosome), five markers (three i(12)(p10), one i(18)(p10), and one pathologic derivative chromosome 15) and one chromoanagenesis (a large number of complex rearrangements at one or multiple chromosomal loci) [36]. Thirteen were diagnosed by CMA and karyotyping, two by CMA only, and 27 by karyotyping only. Nineteen patients were diagnosed based on amniotic fluid (AF), 18 based on CVS, and five had more than one tissue type available for diagnosis. Another 148 samples were classified as unaffected (i.e., no RAA or SUA ≥7 Mb); 39 had microarray and 109 had karyotype. The remaining two samples were found to have feto-placental discrepancy with abnormal cytotrophoblasts and normal fetuses. Mean gestational age was 17.7 weeks (11.0–36.3 weeks).

NIPT analysis of the 192 biobank samples (Cohort A) generated results in 188 samples (Figure 1). Samples could only be run once due to sample constraints; the failure rate was therefore 2.08%. The four failed samples were all from pregnancies where the fetuses had normal cytogenetic results. Of the 188 samples, 150 were low-risk and 38 were high-risk NIPT calls. The average FF was 11.76%. There was no statistical difference between the normal and abnormal samples in the biobank population *(p* = 0.52).

There was one false-positive and five false-negative calls. Sensitivity for detection of RAAs and SUAs ≥7 Mb was 88.1% (37/42; CI 75.00–94.81) and specificity was 99.32% (145/146; CI 96.22–99.88). The one false-positive call was found to be due to confined placental mosaicism type 1 (CPM1, i.e., only in the cytotrophoblast). This sample was reported as dup(11)(p11.12q25); + 13 by NIPT. Karyotype of the direct analysis of cytotrophoblast was 47,XY, +der(13)t(11;13)(q12;q34),t(11;13) but karyotype of a long-term culture of the mesenchyme was 46,XY,t(11;13)(q12;q34).

An overview of the five false-negative results is provided in Table 1; FFs of these samples ranged from 3 to 15%. The rearrangement of Sample 1 consisted of a deletion of part of the band 4p16.3; the size was not studied by array. Although it was visible on the 550 RHG banding karyotype (resolution of 5–10 Mb), the size of the deletion could be between 5 and 7 Mb, which is one possible explanation for the false-negative NIPT result, as this would be below the minimum size threshold for detection by this NIPT assay. For Sample 2, the false-negative result was attributed to the presence of suspected mosaicism, as it is usual in the Palister–Kilian syndrome, resulting from an isochromosome 12p (OMIM 601803). For Sample 3, chromoanasynthesis is the most likely reason for the false-negative result. A biological explanation for the false-negative result for Sample 4 could not be determined; the low FF may explain the discordant result. For Sample 5, the combination of a low FF and the possibility of a mosaicism (isochromosomes have been described as postzygotic events) could explain the false-negative result.

For samples that had a microarray result (15 rearrangements), a good correlation was observed between the segment size of the partial unbalanced rearrangements detected by NIPT and those by microarray, with one exception (Sample 5 in Table 2). NIPT called that sample trisomy 11, while karyotype and array reported a derivative chromosome 11 from a translocation t(11;12) with a partial terminal deletion of chromosome 11 of 2.7 Mb and a partial terminal duplication of trisomy 12 of 11.7 Mb.

Analysis of the 41 deleted/duplicated segments ≥7 Mb in the 35 fetuses with partial unbalanced rearrangements suggest that the detection rate is equivalent for deletions and duplications. By contrast, there may be a higher sensitivity for interstitial rearrangements compared with terminal rearrangements (Table 3); a larger sample size is needed to confirm this trend.

We also wanted to determine a screening strategy to select patients at the highest risk for adverse outcomes, based on MSS risk scores. Lindquist et al. [3] determined prevalence according to the MSS risk score. Using these prevalence values, along with our study specificity of 99.32% and sensitivity of 88.1%, we calculated the expected positive predictive values (PPV) for the different MSS risk scores (Table 4). The theoretical PPV varied from 11 to 64% between the general population and the group with a MSS risk score >1/300.

Cohort B, the referral population cohort (Figure 2), consisted of 3097 samples from women who underwent routine NIPT for common aneuploidies; 88% of these were carried out due to an MSS risk ≥ 1/1000 (Table 5). Mean gestational age was 16.8 weeks (10.1–37.0 weeks). The referral population (Cohort B) had a slightly lower average gestational age than the biobank (Cohort A) samples (16.8 weeks vs. 17.7 weeks, respectively; *p* = 0.003). Gestational age for the biobank cohort ranged from 11.0 to 36.3 weeks. A comparison of the fetal fractions between Cohort A and Cohort B (Table 6) showed a significant difference (*p* < 0.001).

NIPT analysis of the 3097 samples (Cohort B) resulted in 90 (2.9%) first-time failures; the final failure rate is unknown, as samples did not undergo a second-pass test. Of the 3007 samples that received a result (Figure 2), 36 (1.2%) had a positive result for trisomies 13, 18, or 21, and 36 (1.2%) had a positive result for ACAs. In the high-risk population (MSS risk ≥1/1000), 1.09% tested positive for an ACA, and in the first-tier screening population, 3.27% tested positive (Table 7). The 36 ACAs consisted of 10 SUAs, 25 RATs, and one case with multiple anomalies. If we consider only the most significant anomalies, that is, (1) SUAs that do correspond with a classical chromosomal rearrangement, such as deletion, duplication, the translocation derivative chromosome, recombinant of an inversion or marker chromosome and (2) trisomies 8, 9, and 22; trisomies 14 and 15 because of the risk of uniparental disomy; trisomy 16 because of the risk of a poor pregnancy outcome; and trisomy 12 because of the risk of isochromosome 12p (Pallister–Killian syndrome); only 17 samples were positive: 0.57% in the overall study group, 0.53% in the high-risk population with a MSS risk ≥ 1/1000, and 0.65% of the population with first-tier NIPT screening (Table 7).

ACAs, atypical chromosomal anomalies; MSS, maternal serum screen. Group 1 includes the following trisomies: trisomies 8, 9, and 22 because these trisomies often involve the fetus; trisomies 14 and 15 because of their risk for uniparental disomy; trisomy 16 because of the risk of multiple adverse pregnancy outcomes; and trisomy 12 because isochromosome 12p leads to the Pallister–Killian syndrome. Group 2 includes trisomies 3 and 7 as they are known to be frequently confined to the cytotrophoblast [37]. Group 3 includes all other RAAs (trisomies 1, 2, 4, 5, 6, 10, 11, 17, 19, and 20), including trisomies that are non-pathogenic for the fetus such as mosaic trisomy 20, or are generally confined to the placenta.

## 4. Discussion

In this paper, we assessed the ability of a genome-wide NIPT assay to detect the presence of atypical chromosomal anomalies. NIPT was found to be highly specific for their detection, with CPM confirmed as a biological explanation for the single false-positive call. The genome-wide NIPT screen detected the majority of unbalanced rearrangements in the biobank population with a sensitivity of 88.1% (CI 75.00–94.81) and also showed good correlation with array for the detected CNV size. While sensitivity was lower than typically observed for the common trisomies [38], there was a suspected biological explanation for three of the five false-negative results (size of the deletion, suspected mosaicism, and chromoanasyntesis). Due to the high specificity, the number of samples in the referral population that received a positive result for the presence of ACAs was low (1.2%). The failure rate in clinical practice is expected to be lower than that observed in this study because of the ability to run an additional plasma sample if the initial sample fails. The study (biobank) samples were of a slightly higher gestational age than the referral population and, as expected, this was associated with a slightly higher average fetal fraction. However, the difference between the two populations is not clinically meaningful, and thus the performance in the study population is likely similar to what would be observed in a general pregnancy population.

While there is consensus on the value of screening for common fetal aneuploidies, the utility of screening for atypical fetal anomalies is still hotly debated [39] and is still controversial in the clinical community [40]. Even if the adverse prognosis of these ACAs and the utility of a diagnosis is deemed relevant, they are typically considered too rare to be part of a screening policy. In fact, the exact prevalence of these conditions in the population remains uncertain and is probably underestimated because there is not widespread uptake of invasive testing in large unselected pregnancy populations and there are no studies that routinely test all newborns for these abnormalities. In a recent report of French cytogenetic laboratories [31], ACAs represented 12.25% of all prenatal unbalanced chromosomal anomalies. According to Lindquist et al. [3], the overall prevalence is estimated to be 1/1000, not taking into account the potential cases discovered after birth. Moreover, it has been shown that the risk of ACAs increases with the MSS risk score [1,2,3]; therefore, its use as a second-line test would be even more relevant.

Our study shows that genome-wide NIPT is an effective method of screening for atypical fetal anomalies, such as rare trisomies or segmental unbalanced rearrangements, and that it may allow for the detection of a larger number of fetal anomalies without a large increase in unnecessary invasive procedures. Two strategies could contribute to select populations at the highest risk for adverse outcomes: (1) recommending genome-wide NIPT to pregnant women with an elevated risk based on MSS results, as outlined in the Results section above, and (2) screening for a limited number of ACAs known to be associated with a higher risk of adverse outcome. Regarding the first strategy, although the use of genome-wide NIPT to screen for ACAs is possible in a general pregnancy population, we propose restricting offering genome-wide NIPT to pregnant women with an MSS risk score ≥1/1000 (theoretical PPV 32–64%). This recommendation is possible in France because, for singleton pregnancies without a previous history of chromosomal rearrangement, NIPT is a second-tier screening test that is contingent on the MSS risk score. Restricting genome-wide NIPT to pregnant women with a high-risk MSS score could also be a possibility in other countries, such as European Nordic countries, Poland, Romania, Italy, or Australia [41]. Other countries may choose to use genome-wide cfDNA analysis in unselected populations. There is a large variability in how different countries approach NIPT, with some countries recommending it as a first-tier option and others recommending it as part of a contingent model. A recent publication by Benachi et al. [42] described a survey of European healthcare providers highlighted the difference amongst healthcare providers in different countries with regard to offering expanded NIPT options, such as screening for rare autosomal trisomies and copy number variations. It should also be noted that while reimbursement of NIPT at a country level would be a valuable part of prenatal screening, this is currently only possible in certain countries, such as those with national or regional reimbursement programs to fund NIPT. In other countries, patients may be required to pay for the cost of NIPT out-of-pocket, or it may be covered by a patient’s health insurance.

Regarding the second strategy, rare autosomal aneuploidies can be classified into three different groups. Group 1 consists of trisomies most often involving the fetus, including trisomies 8, 9, and 22; trisomies 14 and 15 because of their risk of uniparental disomy; trisomy 16 because of the risk of multiple adverse pregnancy outcomes; and trisomy 12 because isochromosome 12p leads to the Pallister-Killian syndrome. Group 2 includes trisomies known to be most frequently confined to the cytotrophoblast, that is, trisomies 3 and 7 [37]. Group 3 consists of all other RAAs, including trisomies that are non-pathogenic for the fetus, such as mosaic trisomy 20, or are generally confined to placenta. Because a screening test has to detect anomalies with a certain prevalence, we recommend that genome-wide NIPT is used to screen for RAAs in Group 1 only. In our cohort, even if the number is too small to draw definitive conclusions, this strategy is particularly interesting in the low-risk population (first-tier screening) because four of the five anomalies belonged to Groups 2 and 3. If we reanalyze the cohort presented in the recent study of the Dutch NIPT Consortium [28], only 47 women would have received a positive NIPT call for RAAs instead of 101, and all cases with fetal RAAs would have been diagnosed. In our study, targeting SUAs and trisomies 8, 9, 12, 14, 15, 16, and 22 only would generate a positive NIPT call rate of 0.57% (17/3007).

The sensitivity of genome-wide NIPT is not high enough to recommend it to screen fetuses with ultrasound abnormalities, especially given the fact that microdeletions/duplications are frequently involved. We therefore recommend that patients with a fetal structural anomaly observed on ultrasound are offered diagnostic invasive testing with microarray analysis.

One of the main strengths of this study was the availability of samples affected with a chromosome anomaly other than the common trisomies, and a known fetal karyotype for all biobank samples. This allowed determination of the true study’s specificity and sensitivity. We observed a high specificity of 99.32% (CI 96.22–99.88), with only one false-positive result by NIPT that was determined to be due to CPM; CPM is a known biological factor that impacts clinical specificity of NIPT. The availability of a referral population of routine pregnancy samples was another strength. Using this population, and a set strategy, we were able to show that the use of whole-genome NIPT would not result in a substantial increase in unnecessary invasive diagnostic procedures, which is a typical argument against genome-wide NIPT screening. Here, the rate of positive NIPT calls for presence of ACAs did not exceed 1.2%, which is comparable to other studies [33]. Moreover, the positive call rate could be reduced to 0.57% by selecting the most significant ACAs.

One of the main study limitations was that this was a selected set of pregnancy samples and thus may not be truly representative of a larger general pregnancy population. Other recent publications have looked at genome-wide NIPT in patient populations of at least 10,000 patients [5,6,34,43], with a few publications reporting results from populations of over 50,000 patients [28,44,45]. However, the observed assay failure rate and FF distribution in our study population were consistent with those observed in the referral population, suggesting the data is representative. Another limitation of this study was the lack of karyotype and clinical follow-up information for the referral population. Although the aim of the referral population was only to calculate the potential increase in invasive procedures, it could be interesting to have that information to determine the positive and negative predictive value of this genome-wide NIPT assay and to investigate if genome-wide NIPT is cost-effective. A larger study looking at different pregnancy complications could provide further information on which RAAs should be screened for with NIPT. There is also a need for further research to determine whether genome-wide NIPT can be an effective prenatal screening option for detection of rarer fetal anomalies for multiple-gestation pregnancies.

## 5. Conclusions

In conclusion, this study has shown that a genome-wide NIPT assay is an effective non-invasive method of screening for the presence of ACAs in pregnant women. We propose to only screen for the most significant chromosome trisomies and SUAs. In France, we would suggest that genome-wide NIPT screening is considered for all pregnant women with normal ultrasound findings following a high-risk MSS result.

## Figures and Tables

**Figure 1 jcm-09-02466-f001:**
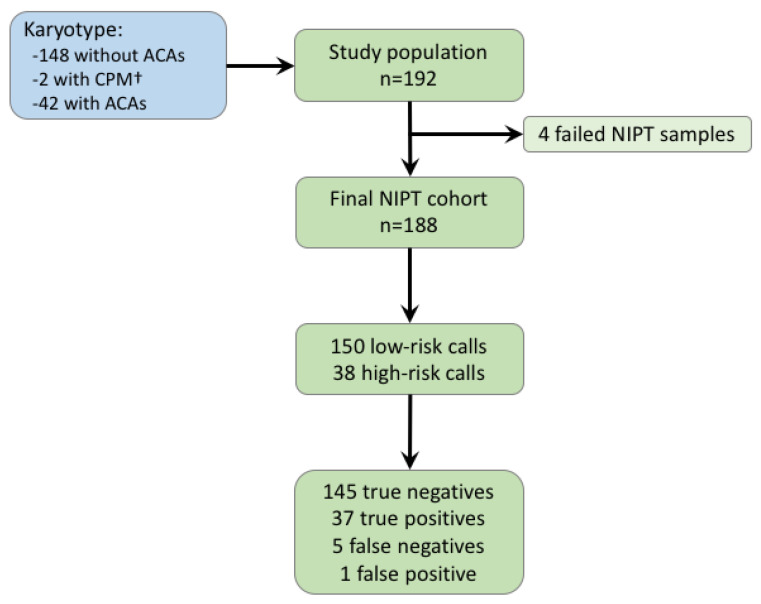
Overview of biobank samples. Overview of the study population, failed samples, and final NIPT cohort. †Two cases had placental discrepancy with abnormal cytotrophoblasts and normal fetuses. The first case was mosaic 46,XY,del(5)(p15)(7)/46,XY(12) by direct cytotrophoblastic karyotype and 46,XY by long-term culture and amniotic fluid culture. The second case was 47,XY, +der(13)t(11;13)(q12;q34),t(11;13) by direct cytotrophoblastic karyotype and 46,XY,t(11;13)(q12;q34) by long-term culture and amniotic fluid culture.

**Figure 2 jcm-09-02466-f002:**
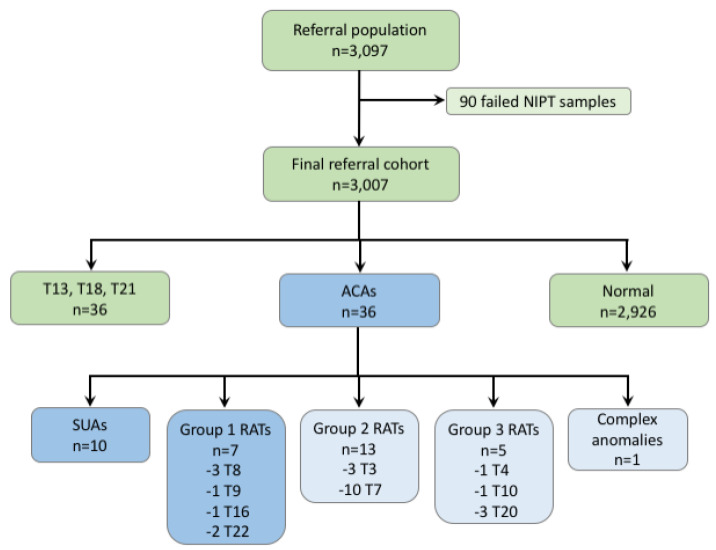
Overview of the referral population, failed samples, and final referral cohort. T, trisomy; ACAs, atypical chromosomal anomalies; SUAs, structural unbalanced anomalies; RATs, rare autosomal trisomies.

**Table 1 jcm-09-02466-t001:** Overview of the five false-negative samples from Cohort A.

Sample	Fetal Fraction	Karyotype	Size (Mb)	Comments
1	10%	46,XX,del(4)(p16.3).ish del(4)(WHS-,D4S3359-)	5–8, based on the karyotype	Possible size < 7 Mb
2	15%	arr[GRCh37] 12p13.33q11(173786_37876500)x3	37.7	Suspected mosaicism
3	9%	46,XX,der(8)?add(8)(p?)?dup(8)(q22q23)dn.ish der(8)(qter->?::?->qter)(D8S504, VIJyRM2053+,wcp8+,VIJyRM2053+).arr[GRCh38] 8p23.3p23.1(158048_6935930)x1,8p23.1p11.23(12585435_38267493)x3, 8p11.22(38314367_39246760)x3, 8p11.22(39247087_39386852)x1,8p11.22(39389765_40264413)x3, 8q22.3q23.2(104688373_111952230)x3, 8q24.3(144972747_146295771)x3	Total deletion = 6.9 Total duplication = 36.2	Chromoanasynthesis
4	3%	46,XX,add(4)(qter).ish add(4)(wcp4-).arr[GRCh37] 5q31.2q35.3(138522878_180715096)x3	41.2	
5	4%	46,XY,i(18)(q10)		

**Table 2 jcm-09-02466-t002:** Correlation of CNV sizes (Mb) between array and NIPT.

Sample	NIPT	Array
1	18.2	12.8
2	9.7	9.8
3	11.5	11.1
4	28.8	29.9
5	Trisomy 11	2.7
6	8.1	6.7
7	10.7	11.5
8	11.3	17.4
9	26	26.6
10	17.2	7.9
11	18.3	18.3
12	9	8.8
13	12.5	12.5
14	13.7	11.8
15	60.1	59.9

CNV, copy number variation; NIPT, non-invasive prenatal test.

**Table 3 jcm-09-02466-t003:** Unbalanced structural rearrangements observed in study samples.

Type of Rearrangement	Observed on Karyotype, n	Detected by NIPT, n	Detection Rate, % (95% CI)
Deletion	13	11	84.6 (54.6–98.1)
Duplication	28	24	85.7 (67.3–96.0)
Interstitial	5	5	100 (47.8–100)
Terminal	36	28	77.8 (60.9–89.9)

CI, confidence interval.

**Table 4 jcm-09-02466-t004:** PPVs based on the prevalence of ACAs and the MSS risk score.

Measurement	General Population	MSS Score 1/51–1/1000	MSS Score >1/1000	MSS Score 1/51–1/300	MSS Score >1/300
Prevalence ^1^	0.10%	0.37%	0.61%	1.01%	1.40%
PPV	11%	32%	44%	56%	64%

^1^ Values are based on study by Lindquist et al. [3]. PPV, positive predictive value; ACAs, atypical chromosomal anomalies; MSS, maternal serum screen.

**Table 5 jcm-09-02466-t005:** NIPT indications for the referral population.

Population Type	Failed	No Anomalies	Common Trisomies	ACAs	Total, n (%)
MSS ≥ 1/1000	71	2596	35	29	2731 (88)
MSS < 1/1000	9	139	0	1	149 (5)
Parental Robertsonian translocation	0	2	0	0	2 (0)
Previous history of fetal trisomy	2	51	0	1	54 (2)
First-tier screening	8	147	1	5	161 (5)
**Total**	90	2935	36	36	3097

ACAs, atypical chromosomal anomalies; MSS, maternal serum screen.

**Table 6 jcm-09-02466-t006:** Fetal fractions of the biobank and referral populations.

Measurement	Biobank Samples (Cohort A) (n = 189)	Normal Biobank Samples from Cohort A (n = 147)	Abnormal Biobank Samples from Cohort A (n = 42)	Referral Samples (Cohort B) (n = 3007)
Average	12.27%	12.40%	11.76%	11%
Median	11%	11%	10.5%	10%
Range	3–35%	4–35%	3–24%	2–35%

**Table 7 jcm-09-02466-t007:** ACAs in the referral population according to NIPT indication.

Population Type	Total, Excluding Failures	Group 1 + SUAs	Group 2	Group 3	Prevalence of All ACAs	Prevalence of Group 1 + SUAs
MSS ≥ 1/1000	2660	14	10	5	1.09%	0.53%
MSS < 1/1000	140	1	0	0	0.71%	0.71%
First-tier screening	153	1	3	1	3.27%	0.65%
Previous history of fetal trisomy or parental Robertsonian translocation	54	1	0	0	1.85%	1.85%

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
