# Peer review of "Strategy for Use of Genome-Wide Non-Invasive Prenatal Testing for Rare Autosomal Aneuploidies and Unbalanced Structural Chromosomal Anomalies"

_jcm, 2020, doi:10.3390/jcm9082466_

Round 1
Reviewer 1 Report
In the present study authors have showed that a genome-wide NIPT assay is an effective noninvasive method of screening for presence of atypical chromosomal anomalies in pregnant women. Based on the study they propose to only screen for the most significant chromosome trisomies and structural unbalanced anomalies.
The study is of significant importance and planned and performed well. Authors have discussed the limitations adequately, however, it will be valuable to include and discuss why the failed NIPT tests could not be repeated. Was that due to sample constraints or procedural or technical limitation?
Author Response
Point 1: The study is of significant importance and planned and performed well. Authors have discussed the limitations adequately, however, it will be valuable to include and discuss why the failed NIPT tests could not be repeated. Was that due to sample constraints or procedural or technical limitation?
Response 1: Samples could only be run once due to sample constraints. We have updated the text to reflect this (Page 4 Line 167). With normal clinical samples, not biobank samples of limited volume, samples could be rerun using residual plasma.
Reviewer 2 Report
In this manuscript, Kleinfinger, et al. have carried out a screening for a genome-wide noninvasive prenatal test (NIPT) to detect atypical chromosomal anomalies (ACAs) using two sample French cohorts. They conclude that genome-wide NIPT is an effective noninvasive method of screening for presence of ACAs in pregnant women. The authors also suggest that in France, the genome-wide NIPT screening should be considered for all pregnant women with normal ultrasound findings, following a high-risk maternal serum screening result. Genome wide-NIPT has been previously used in different countries and results from large cohorts have already been reported. Although incremental to the field, the manuscript reports valuable experimental data that may serve to optimize the ongoing implementation of this technique. However, some concerns must be addressed before publication.
Specific comments
- Noninvasive prenatal testing has been widely adopted throughout Europe, Australia and the USA, but only a few countries/states have a national policy on the use of NIPT. Therefore, the variation in NIPT utilization is considerable. While the authors discuss the implications of the Results in the context of the French Healthcare system, they should also discuss the existence of the large variability among different countries.
- Experimental section. If available, when describing the cohort it would be good to know if it contains samples from women that underwent in-vitro fertilization (IVF) procedures. If so, please provide the corresponding data.
- Introduction/Discussion. The authors may wish to enrich these sections by citing and commenting the following recent articles:
Teder et al. Computational framework for targeted high-coverage sequencing based NIPT. PLoS One. 2019 Jul 8;14(7):e0209139. doi: 10.1371/journal.pone.0209139.
Gadsbøll et al. Current use of noninvasive prenatal testing in Europe, Australia and the USA: A graphical presentation. Acta Obstet Gynecol Scand. 2020 Jun;99(6):722-730. doi: 10.1111/aogs.13841.
Benachi et al. Understanding attitudes and behaviors towards cell-free DNA-based noninvasive prenatal testing (NIPT): A survey of European health-care providers. Eur J Med Genet. 2020 Jan;63(1):103616. doi: 10.1016/j.ejmg.2019.01.006.
- Page 10, lines 327 and 328. “One of the main study limitations was that this was a selected set of pregnancy samples and thus may not be truly representative of a larger general pregnancy population.” Here, the authors properly emphasize the limited number of samples analyzed. This comment should be expanded in the Discussion section by comparing the actual numbers of this study with those of larger cohorts.
- Is genome-wide NIPT cost-efficient?
Minor comments
- Page 1, lines 41 and 42. “The rate of positive results indicating an anomaly 41 was 1.2% (36/3,007) for all positive results and 0.57% (17/3,007)…”. This sentence is confusing due to the redundant use of the word “positive”. Please rephrase.
- Page 3, lines 129, 130. The abbreviation cfDNA (circulating free DNA? cell-free fetal DNA?) should be spelled out the first time it is mentioned
- Page 4, line 161. Please spell out “mos”
- Page 4, line 161. “karyotype” instead of “karyotye”
- Supplementary Table 1. The abbreviations in the table legend should be listed in alphabetical order to facilitate the reading.
Author Response
Specific Comments Point 1: Noninvasive prenatal testing has been widely adopted throughout Europe, Australia and the USA, but only a few countries/states have a national policy on the use of NIPT. Therefore, the variation in NIPT utilization is considerable. While the authors discuss the implications of the Results in the context of the French Healthcare system, they should also discuss the existence of the large variability among different countries.
Specific Comments Response 1: We have added this discussion point to Page 9 Lines 302–314.
Specific Comments Point 2: Experimental section. If available, when describing the cohort it would be good to know if it contains samples from women that underwent in-vitro fertilization (IVF) procedures. If so, please provide the corresponding data.
Specific Comments Response 2: This data is not available so cannot be added to the manuscript.
Specific Comments Point 3: Introduction/Discussion. The authors may wish to enrich these sections by citing and commenting the following recent articles:
Teder et al. Computational framework for targeted high-coverage sequencing based NIPT. PLoS One. 2019 Jul 8;14(7):e0209139. doi: 10.1371/journal.pone.0209139.
Gadsbøll et al. Current use of noninvasive prenatal testing in Europe, Australia and the USA: A graphical presentation. Acta Obstet Gynecol Scand. 2020 Jun;99(6):722-730. doi: 10.1111/aogs.13841.
Benachi et al. Understanding attitudes and behaviors towards cell-free DNA-based noninvasive prenatal testing (NIPT): A survey of European health-care providers. Eur J Med Genet. 2020 Jan;63(1):103616. doi: 10.1016/j.ejmg.2019.01.006.
Specific Comments Response 3: Thank you for the suggested references. We have included Gadsbøll et al. on Page 9 Line 304 and Benachi et al. on Page 9 Line 307.
Specific Comments Point 4: Page 10, lines 327 and 328. “One of the main study limitations was that this was a selected set of pregnancy samples and thus may not be truly representative of a larger general pregnancy population.” Here, the authors properly emphasize the limited number of samples analyzed. This comment should be expanded in the Discussion section by comparing the actual numbers of this study with those of larger cohorts.
Specific Comments Response 4: We have updated the text to address this comment as follows (Page 10 Lines 346–348): “Other recent publications have looked at genome-wide NIPT in patient populations of at least 10,000 patients [5,6,34,43], with a few publications reporting results from populations of over 50,000 patients [28,44,45].”
Specific Comments Point 5: Is genome-wide NIPT cost-efficient?
Specific Comments Response 5: We have added this as an area for further research (see Page 10 Line 354–355).
Minor Comments Point 1: Page 1, lines 41 and 42. “The rate of positive results indicating an anomaly 41 was 1.2% (36/3,007) for all positive results and 0.57% (17/3,007)…”. This sentence is confusing due to the redundant use of the word “positive”. Please rephrase.
Minor Comments Response 1: We have updated the text as follows (Page l Lines 41–43): “The rate of positive results indicating an anomaly was 1.2% (36/3,007) and 0.57% (17/3,007) when limited to significant unbalanced chromosomal anomalies and trisomies 8, 9, 12, 14, 15, 16, and 22.”
Minor Comments Point 2: Page 3, lines 129, 130. The abbreviation cfDNA (circulating free DNA? cell-free fetal DNA?) should be spelled out the first time it is mentioned
Minor Comments Response 2: We have updated the text to “cell-free (cf)DNA extraction step” (Page 3 Line 130)
Minor Comments Point 3: Page 4, line 161. Please spell out “mos”
Minor Comments Response 3: We have updated the text to spell out “mos” as follows (Page 4 Line 162): “The first case was mosaic 46,XY,del(5)(p15)[7]/46,XY[12]”
Minor Comments Point 4: Page 4, line 161. “karyotype” instead of “karyotye”
Minor Comments Response 4: We made the correction (Page 4 Line 162).
Minor Comments Point 5: Supplementary Table 1. The abbreviations in the table legend should be listed in alphabetical order to facilitate the reading.
Minor Comments Response 5: We have updated the table legend of Supplementary Table 1 as requested.
Reviewer 3 Report
Abstract should be corrected to be more coherent. It does not clearly present neither purpose of the study, method, nor the correlation between results and conclusions. Dividing abstract into separate sections would add value to its content and would make it more understandable.
Please explain why patients did not have the possibility of access to your results of the karyotype analysis - by agreeing to take part in your study, they should have been given the possibility of knowing the results and changing their mind about their decision. What was the procedure in such cases? I understand that it was the design of your study, but still it gives me some ethical concern.
Specificity of 88.1% an acceptable for such type of testing and cannot be considered high, merely acceptable as it influences important decisions about the following course of pregnancy.
Even though introducing general country-reimbursed NIPT screening would be a valuable part of prenatal diagnosing, it must be underlined that it is possible only in selected countries.
Author Response
Point 1: Abstract should be corrected to be more coherent. It does not clearly present neither purpose of the study, method, nor the correlation between results and conclusions. Dividing abstract into separate sections would add value to its content and would make it more understandable.
Response 1: Author guidelines for the Journal of Clinical Medicine require abstracts to be a single paragraph without any headings so we cannot divide it into separate sections as suggested by the reviewer.
Point 2: Please explain why patients did not have the possibility of access to your results of the karyotype analysis - by agreeing to take part in your study, they should have been given the possibility of knowing the results and changing their mind about their decision. What was the procedure in such cases? I understand that it was the design of your study, but still it gives me some ethical concern.
Response 2: Patients in Cohort A with karyotype results based on invasive testing did so as part of their standard clinical care and received those results. For patients in the referral population, genome-wide screening was at that time a controversial issue and patients consented to be part of a non-interventional study only. We have updated the text to help clarify this (Page 3, Lines 116-119): “Genome-wide NIPT analysis was subsequently performed for this study. As patients consented to a non-interventional study only, results of the genome-wide analysis were not reported, and we did not contact them to obtain further follow-up information regarding fetal karyotypes and clinical outcomes.”
Point 3: Specificity of 88.1% an acceptable for such type of testing and cannot be considered high, merely acceptable as it influences important decisions about the following course of pregnancy.
Response 3: We have updated the text to reflect this comment as follows (Page 1 Lines 44–45): “These results show that genome-wide NIPT can screen for ACAs with an acceptable sensitivity and a small increase in invasive testing,..”
Point 4: Even though introducing general country-reimbursed NIPT screening would be a valuable part of prenatal diagnosing, it must be underlined that it is possible only in selected countries.
Response 4: We have added this discussion point to Page 9 Lines 310–314.
Reviewer 4 Report
General Comments:
The manuscript deals with “Strategy for use of genome-wide noninvasive prenatal testing for rare autosomal aneuploidies and unbalanced structural chromosomal anomalies” The authors utilized two study cohorts, namely, A and B. Cohort consisted of banked plasma samples and the B is the referral population who already had NIPT.
This is an interesting study that potentially helps in the development of a strategy for laboratory diagnosis of rare autosomal aneuploidies and unbalanced structural chromosomal anomalies in prenatal screening. The authors conclude that a genome-wide NIPT assay is an effective noninvasive method of screening for presence of ACAs in pregnant women. We propose to only screen for the most significant chromosome trisomies and SUAs. However, the goal of this study was focused in the application of the strategy in France. The reviewer would like the authors to include the pro and cons of introducing it in USA and other countries. The manuscript is well written and organized.
Specific Comments:
Page 1-
Line 33- Atypical fetal anomalies --- please explain if these are chromosomal or ultrasound
Page 2-
Line 56- rare autosomal aneuploidies (RAAs) and or structural unbalanced anomalies (SUAs) … Either give a reference to support or expand it to explain
Line 61 - common trisomies …. Mention these in parenthesis.
Line 62 -
Page 3
Line 118 - outcome information were not available for patients in the referral population…… please give a reason
Line 123 - At least 15 mitoses…. Or metaphases? Clarify
Line 123 - Microarrays were performed …. Clarify if this is on all cases?
Line 148 - (Supplementary Table 1)…. This is not the correct table 1, explain.
Page 4
Line 149: placenta (type III) …. Either give a reference to support or explain.
Line 168…… . The average FF - describe FF used as first time in the text
Line 172…. placental mosaicism type 1 (CPM1). Either give a reference to support or explain.
Page 4 and 5-
Line 177-179…. Although it was visible on the 550 RHG banding karyotype (resolution
of 5–10 Mb), the size of the deletion could be between 5–7 Mb and this would explain the false-negative NIPT result……..This need to be supported by evidence or else delete the speculation.
Line 180…… For sample 2, the false-negative result was attributed to the presence of suspected mosaicism . …. Provide the Log2 ratio for support.
Line 191…..with a median of 17.8 Mb and a range of 7.9–72 Mb….Explain by which technique?
Line 195…… 11 from a translocation t(11;12) with a deletion of chromosome 11 of 2.7Mb and a trisomy 12 …. Clarfiy if you meant partial deletion and partial trisomy 12
Page 8
Line 266 ….. biological explanation……….. give examples of biological explanation related to the study samples
Line 270…(biobank) samples were of a slightly higher gestational age than the referral population…… ( include the range)
Page 9
Line 287…. anomalies … this refers which type, explain
Author Response
General Comments Point 1: This is an interesting study that potentially helps in the development of a strategy for laboratory diagnosis of rare autosomal aneuploidies and unbalanced structural chromosomal anomalies in prenatal screening. The authors conclude that a genome-wide NIPT assay is an effective noninvasive method of screening for presence of ACAs in pregnant women. We propose to only screen for the most significant chromosome trisomies and SUAs. However, the goal of this study was focused in the application of the strategy in France. The reviewer would like the authors to include the pro and cons of introducing it in USA and other countries. The manuscript is well written and organized.
General Comments Response 1: We have added this discussion point to Page 9 Lines 302–314.
Specific Comments Point 1: Page 1-Line 33- Atypical fetal anomalies --- please explain if these are chromosomal or ultrasound
Specific Comments Response 1: We have updated the text to address this comment as follows (Page 1 Line 33): “Atypical fetal chromosomal anomalies…”
Specific Comments Point 2: Page 2- Line 56- rare autosomal aneuploidies (RAAs) and or structural unbalanced anomalies (SUAs) … Either give a reference to support or expand it to explain
Specific Comments Response 2: This sentence was previously referenced using the following three references (Page 2 Lines 56–57):
- Huang, S.; Chang, C.; Cheng, P.; Hsiao, C.; Soong, Y.; Duan, T. First-trimester combined screening is effective for the detection of unbalanced chromosomal translocations at 11 to 12 weeks of gestation. Reprod. Sci. 2014, 21, 594-600, doi:10.1177/1933719113508818
- Torring, N.; Petersen, O.B.; Becher, N.; Vogel, I.; Uldbjerg, N. First trimester screening for other trisomies than trisomy 21, 18, and 13. Prenat. Diagn. 2015, 35, 612-619, doi:10.1002/pd.4584.
- Lindquist, A.; Poulton, A.; Halliday, J.; Hui, L. Prenatal diagnostic testing and atypical chromosome abnormalities following combined first-trimester screening: implications for contingent models of non-invasive prenatal testing. Ultrasound Obstet. Gynecol. 2018, 51, 487-492, doi:10.1002/uog.18979
Specific Comments Point 3: Page 2-Line 61-62 - common trisomies …. Mention these in parenthesis.
Specific Comments Response 3: We mentioned what the common trisomies were earlier in this paragraph (Page 2 Line 54) so don’t think it is necessary to call it out again.
Specific Comments Point 4: Page 3 Line 118 - outcome information were not available for patients in the referral population…… please give a reason
Specific Comments Response 4: Patients in the referral population consented to a non-interventional study only. This is the reason why results from the genome-wide NIPT analysis were not reported to the patients and why we did not contact them at a later date to collect outcome information. We have updated the text to reflect this as follows (Page 3, Lines 117–120): “Genome-wide NIPT analysis was subsequently performed for this study. As patients consented to a non-interventional study only, results of the genome-wide analysis were not reported and, we did not contact them to obtain further follow-up information regarding fetal karyotypes and clinical outcomes.”
Specific Comments Point 5: Page 3 Line 123 - At least 15 mitoses…. Or metaphases? Clarify
Specific Comments Response 5: We have updated the text to clarify this as follows (Page 3 Line 124): “At least 15 metaphases on two cultures were analyzed.”
Specific Comments Point 6: Page 3 Line 123 - Microarrays were performed …. Clarify if this is on all cases?
Specific Comments Response 6: As outlined earlier in the Methods (Page 3 Line 102), we noted that patients in Cohort A had undergone either karyotyping or a microarray. In addition, we also call out the number of samples that underwent microarray vs karyotyping in the Results section (Page 4 Lines 154–155). We therefore do not think it is necessary to add this detail again.
Specific Comments Point 7: Page 3 Line 148 - (Supplementary Table 1)…. This is not the correct table 1, explain.
Specific Comments Response 7: This is the correct table. An overview of the 42 cases where an RAA or SUA ≥7Mb was found at invasive testing is provided in Supplementary Table 1 (Page 4 Line 149).
Specific Comments Point 8: Page 4 Line 149: placenta (type III) …. Either give a reference to support or explain.
Specific Comments Response 8: We have updated the text to clarify this as follows (Page 4 Line 150–151): “three trisomy 16 confined to the placenta (type III, i.e. found in both the cytotrophoblast and in the mesenchyme)”
Specific Comments Point 9: Page 4 Line 168…… . The average FF - describe FF used as first time in the text
Specific Comments Response 9: FF (fetal fraction) was previously described on Page 2 Line 60.
Specific Comments Point 10: Page 4 Line 172…. placental mosaicism type 1 (CPM1). Either give a reference to support or explain.
Specific Comments Response 10: We have updated the text to clarify this (Page 4 Line 174–175): “The one false-positive call was found to be due to confined placental mosaicism type 1 (CPM1, i.e., only in the cytotrophoblast).”
Specific Comments Point 11: Page 4 and 5- Line 177-179…. Although it was visible on the 550 RHG banding karyotype (resolution of 5–10 Mb), the size of the deletion could be between 5–7 Mb and this would explain the false-negative NIPT result……..This need to be supported by evidence or else delete the speculation.
Specific Comments Response 11: As the size was not studied by array, we cannot say for certain what the actual size of the deletion for this sample is. We have updated the text to further clarify that the explanation that we provide is only one possible explanation for the false-negative result as follows (Page 5 Line 180–183): “Although it was visible on the 550 RHG banding karyotype (resolution of 5–10 Mb), the size of the deletion could be between 5–7 Mb which is one possible explanation for the false-negative NIPT result as this would be below the minimum size threshold for detection by this NIPT assay.”
Specific Comments Point 12: Page 4 and 5- Line 180…… For sample 2, the false-negative result was attributed to the presence of suspected mosaicism . …. Provide the Log2 ratio for support.
Specific Comments Response 12: As this sample was a segmental case and not a full aneuploidy we are not able to provide a Log2e of the LLR score. We suspected that this was a mosaic sample based on the usual cytogenetics of Palister-Killian syndrome. We have updated the text to clarify this (Page 5 Line 183-185): “. For sample 2, the false-negative result was attributed to the presence of suspected mosaicism as is usual in Palister-Kilian syndrome resulting from an isochromosome 12p (OMIM 601803).”
Specific Comments Point 13: Page 4 and 5 Line 191…..with a median of 17.8 Mb and a range of 7.9–72 Mb….Explain by which technique?
Specific Comments Response 13: We have deleted that sentence as we think it may be confusing to the reader.
Specific Comments Point 14: Page 4 and 5 Line 195…… 11 from a translocation t(11;12) with a deletion of chromosome 11 of 2.7Mb and a trisomy 12 …. Clarfiy if you meant partial deletion and partial trisomy 12
Specific Comments Response 14: We have updated the text to clarify this as follows (Page 5 Lines 197–199): “…11 from a translocation t(11;12) with a partial terminal deletion of chromosome 11 of 2.7Mb and a partial terminal duplication of trisomy 12 of 11.7Mb"
Specific Comments Point 15: Page 8 Line 266 ….. biological explanation……….. give examples of biological explanation related to the study samples
Specific Comments Response 15: We have updated the text to address this comment as follows (Page 8 Lines 270–271): “….biological explanation for three of the five false-negative results (size of the deletion, suspected mosaicism, and chromoanasyntesis).”
Specific Comments Point 16: Page 8 Line 270…(biobank) samples were of a slightly higher gestational age than the referral population…… ( include the range)
Specific Comments Response 16: We have added the mean gestational age and range for the biobank samples to the Results section (Page 6 Lines 224–227) as follows: “The referral population (Cohort B) had a slightly lower average gestational age than the biobank (Cohort A) samples (16.8 weeks vs 17.7 weeks, respectively; p=0.003). Gestational age for the biobank cohort ranged from 11.0–36.3 weeks”
Specific Comments Point 17: Page 9 Line 287…. anomalies … this refers which type, explain
Specific Comments Response 17: We have updated the text to reflect this as follows (Page 9 Lines 291–292): “…atypical fetal anomalies such as rare trisomies or segmental unbalanced rearrangements”